# Perception of Religious Brothers and Sisters and Lay Persons That Prayers Go Unanswered Is a Matter of Perceived Distance from God

**Arndt Büssing** [1,2,*], **Stephan Winter** [3] **and Klaus Baumann** [4]

[1] Professorship Quality of Life, Spirituality and Coping, Faculty of Health, Witten/Herdecke University, 58313 Herdecke, Germany

[2] IUNCTUS—Competence Center for Christian Spirituality, Philosophical-Theological Academy, 48149 Münster, Germany

[3] Faculty of Catholic Theology, Eberhard Karls University, 72074 Tübingen, Germany; stephan.winter@uni-tuebingen.de

[4] Caritas Science and Christian Social Work, Faculty of Theology, Albert-Ludwig University, 79085 Freiburg, Germany; klaus.baumann@theol.uni-freiburg.de

\* Correspondence: arndt.buessing@uni-wh.de

**Abstract:** Background: Sometimes prayer life can be difficult even for very religious persons, who may experience phases of "spiritual dryness", which may have a negative effect on their well-being. Methods: To address this topic, we analyzed three contrasting groups of persons (religious brothers and sisters (RBS), $n = 273$; Catholic lay persons (CLP), $n = 716$; other lay persons (OLP), $n = 351$) with standardized measures and investigated how often indicators of spiritual dryness were perceived within these groups and how the perception that private prayers go unanswered could be a result of this. Results: Spiritual dryness was highest in RBS compared to RLP and OLP. For RBS, perception of being "spiritually empty" was the best predictor of prayers going unanswered, indicating emotional/spiritual exhaustion, while in OLP, the perception that God is "distant" was the best predictor, indicating that, particularly in this (younger) group, spiritual doubt is of particular relevance. For CLP, feeling that God is distant, feeling abandoned by God, and feeling "spiritually empty" were similarly relevant predictors of feelings that prayers go unanswered. Conclusions: This knowledge may help psychologists/psychotherapists, pastoral workers, and spiritual advisors to differentiate the underlying causes of spiritual dryness (in terms of "discernment") and thus support persons struggling with God, their faith, and life.

**Keywords:** prayer; unanswered prayers; spiritual dryness; distance from God; emotional exhaustion; spiritual emptiness; well-being; religious brothers and sisters; lay persons

---

## 1. Introduction

When persons who are supposed to live a vital prayer life, such as priests and religious brothers and sisters, are confronted with religious doubts or struggles, their basic foundation in life (their faith and belief in God) may be substantially impaired. As a matter of fact, and not only in a Christian context, such existential doubts and struggles are expected to be necessary ingredients of an authentic life of faith, although these may be experienced as deeply disturbing or confusing. One indicator of such struggles might be that persons perceive that their private prayers go "unanswered" and that it makes no sense to pray anymore because God is perceived as distant, not listening, and not caring, or that they have been completely abandoned by God (Büssing et al. 2013, 2016a, 2017a). These indicators of "spiritual dryness" are often connected, but not necessarily as a sequence. It might be that persons



perceive that their prayers are in vain, but they nevertheless long for a closer relation with God, who is perceived as distant. On the other hand, the feeling of being ignored by God may result in an emotional/spiritual withdrawal, resulting in a stopping of private prayer life. Emotional exhaustion and spiritual "emptiness" would thus be considered either as further negative results or as the consequences of a perceived intense or long-lasting distance from God.

Perceived abandonment by God, spiritual struggles, desolation, and "desert experiences" in a person's spiritual life are classic themes in religious literature (Plattig and Bäumer 2010; Plattig 1994; Büssing and Dienberg 2019), as well as in psychology/health research (Exline 2013; Büssing et al. 2013). These experiences were not only felt by some extraordinary saints of the past (Weismayer 2019; Höffner 2019), but also by today's persons who are committed to living a religiously informed life, including routines and special moments of more or less intrinsically motivated prayer in its multiple forms. Phases of "spiritual dryness" (either long lasting or short) were perceived occasionally by 50% of non-ordained Catholic pastoral workers and often or even regularly by 13% (Büssing et al. 2016a). In Catholic priests, these phases were experienced by 46% occasionally and by 12% often or regularly (Büssing et al. 2017a). Relevant predictors of spiritual dryness were low perception of the sacred in their life, low sense of coherence, depressive states, and emotional exhaustion (Büssing et al. 2016a, 2017a). Among the different strategies to cope with these phases, priests also used explicit spiritual practices (i.e., personal prayer, meditation) (Büssing et al. 2017b). A more intense prayer life (as a "therapy"), as recommended by some persons (De Mello 2005, p. 160), is difficult when one perceives that prayers are not "noticed" and responded to by God. However, prayer as a ritual or as a matter of constancy, faithfulness, and endurance is also a coping mechanism.

The intensity of prayer life and the function of prayer in daily life activity might be different in specific groups of persons. Religious brothers and sisters, for example, are expected to pray several times a day (i.e., the mandatory Liturgy of Hours); priests, in addition to the Liturgy of Hours, use formal forms of prayers during religious ceremonies and may pray privately as well; religious lay persons may pray privately in different life situations (in times of illness, to give thanks and praise, etc.). Private prayers might be either formal or less formal and rather spontaneous in wording and form. Further, prayers could be differentiated as petitionary, colloquial, ritualist, or meditative (Poloma and Pendleton 1991). In most cases, prayers are an indicator of a "relation" with God, which might be either vital or a ritual only. In Catholic priests and also non-ordained Catholic pastoral workers, it was shown that private prayers were moderately related to the perception of the sacred in life, while the mandatory Liturgy of Hours was weakly or only marginally associated (Büssing et al. 2016b). This would indicate that, rather, spontaneous prayer is related to a vital perception of the sacred in life, while the mandatory prayer is often perceived as a duty and not necessarily an indicator of a vital, existential relation to God.

Of course, the question of whether prayers indeed go "unanswered" is first of all a matter of expectation. Praying means to be in contact with God, to communicate with Him. Whether and how the "responses" are perceived is a different question. One may not notice the "response", the expectations of how God should react may be very specific, or one may expect an immediate response that might not be given yet. It is also true that this specific expecting form of prayer is an attempt to control God. A different form of prayer is meditative or reflecting prayer, which means to be with God without any specific expectation, just being in resonance whatever may appear—even when it is silence, which may nevertheless calm the soul (and might be an answer, too).

Nevertheless, when do persons perceive that their prayer life and relation with God become difficult? How would this affect the well-being of these persons and their perception of their prayers? To address these questions, we investigated three contrasting groups of persons: (1) religious brothers and sisters who are expected to pray several times a day, (2) catholic lay persons who were tasked by their bishop to preside over the "Liturgy of the Word" on Sundays (and/or during the week) and are thus supposed to pray, and (3) other lay persons from other different professions. Within these groups, we investigated how often indicators of spiritual dryness were perceived and how the

perception that private prayers go unanswered could be predicted. This knowledge might be of relevance not only theoretically (i.e., for theology, psychology, and social science), but also practically for psychologists/psychotherapists, pastoral workers, and spiritual advisors.

## 2. Materials and Methods

### *2.1. Enrolled Persons*

The participants were recruited in three consecutive cross-sectional studies. They were informed about the purpose of the study and assured of confidentiality and their right to withdraw their participation at any time. By completing the anonymous online questionnaire, the participants agreed that their data would be evaluated anonymously.

(1) Religious brothers and sisters (RBS) were invited either directly by email to their respective congregations to take part in an anonymous online survey or by participation calls that were sent to the German congregation superiors ("Ordens-Oberen-Konferenz") to be forwarded. Among the participants (both male and female), 71% were from Franciscan orders and 29% from other religious congregations (among these, 30% Benedictines, 16% Jesuits, 10% Schönstadt Fathers, 4% Dominicans, etc.).

(2) Catholic lay persons (CLP), commissioned by their bishop to preside over the "Liturgy of the Word" at Sundays (or during the week) were invited by their respective dioceses to participate. These were pastoral assistants or parish expert workers (20%) or voluntary lay persons (80%). They had presided over the "Liturgy of the Word" for 31.5 ± 13.7 years [1–74]). Among them, 56% pray privately at a regular level, 35% pray often, and 7% seldom pray. At any rate, it seems, these lay persons can be considered to be more or less securely religiously attached individuals (Granqvist 2014; Heuft 2016), or it can be considered that their religious prayer is motivationally central for them (Huber and Huber 2012).

(3) Other lay persons (OLP) not professionally affiliated with the Catholic church were from other different professions (i.e., 15% health professionals and psychologists, 27% teachers, 26% students, and 32% other); they were recruited via snowball sampling. Most of them were Catholics (52%), 34% were Protestants, 5% were of other religious denominations, and 8% had no specific religious denomination.

### *2.2. Measures*

#### 2.2.1. Spiritual Dryness

To operationalize feelings of "spiritual dryness", we used the six-item Spiritual Dryness Scale (SDS), which had good internal consistency (Cronbach's $\alpha$ = 0.87) (Büssing et al. 2013). The instrument addresses whether or not individuals have experienced phases of "spiritual dryness", including feeling that God is distant, feeling that one's prayers go unanswered, feeling "spiritually empty" or not able to give any more (in terms of spiritual exhaustion), and, finally, feelings of being abandoned by God. The items of this instrument were formulated in order to fit into the daily life experiences of religious individuals. Response options on a Likert scale were "not at all" (1), "rarely" (2), "occasionally" (3), "fairly often" (4), and "regularly" (5). The SDS scores are mean scores and represent the perceived lack/shortage.

#### 2.2.2. Daily Spiritual Experiences

The Daily Spiritual Experience Scale (DSES) was developed as a measure of a person's perception of the transcendent in daily life, and thus, the items measure experience rather than particular beliefs or behaviors (Underwood and Teresi 2002; Underwood 2011). Here, we used the six-item version (DSES-6; alpha = 0.91), which uses specific items, such as feeling God's presence, feeling God's love, a desire to be closer to God (union), finding strength/comfort in God, and being touched by the beauty of creation

(Underwood and Teresi 2002). The response categories on a Likert scale are "many times a day", "every day", "most days", "some days", "once in a while", and "never/almost never". Item scores are mean scores.

### 2.2.3. Stude and Awe (GrAw-7)

To measure feelings of awe and subsequent feelings of gratitude, we used the seven-item Gratitude/Awe (GrAw-7) scale (Büssing et al. 2018). This scale addresses the "emotional response to an immediate and 'captured' experience, and not an emotional response in response to goodwill of a person" (Büssing et al. 2018). Examples of items are the following: "In certain places, I become very quiet and devout"; "I stop and am captivated by the beauty of nature"; "I pause and stay spellbound at the moment"; "I stop and then think of so many things for which I'm really grateful". The internal reliability of these items is good (Cronbach's alpha = 0.83). All items were evaluated on a four-point scale (0—never; 1—seldom; 2—often; 3—regularly). The results were sum scores ranging from 0 to 21.

### 2.2.4. Well-Being Index (WHO-5)

To assess participants' well-being, we used the WHO-Five Well-being Index (WHO-5). This short scale avoids symptom-related or negative phrasings and measures of well-being instead of absence of distress (Bech et al. 2003). Representative items are "I have felt cheerful and in good spirits" or "My daily life has been filled with things that interest me". Respondents assess how often they had the respective feelings within the last two weeks, ranging from "at no time" (0) to "all of the time" (5). Here, we report the sum scores.

### 2.3. Statistical Analyses

Descriptive statistics as well as first order correlations (Spearman rho) and regression analyses (inclusion models) were computed with SPSS 23.0. The significance level was set at $p < 0.01$. With respect to classifying the strength of the observed correlations, we regarded $r > 0.5$ as a strong correlation, $r$ between 0.3 and 0.5 as a moderate correlation, $r$ between 0.2 and 0.3 as a weak correlation, and $r < 0.2$ as negligible or no correlation.

## 3. Results

### 3.1. Description of the Sample

The proportion of women and men and age was significantly different in the three study samples, with the oldest in RBS and the youngest in OLP (Table 1). Women predominated in OLP. Their well-being scores and their perception of awe/gratitude were similar, while spiritual dryness and perception of the sacred were highest in RBS (Table 1).

### 3.2. Perceptions of God and Praying within the Three Samples

A total of 9% of OLP perceived that own prayers go unanswered often/regularly, and 24% perceived that this is the case sometimes; 4% of CLP perceived it often/regularly, and 23% perceived it sometimes; 8% of RBS (which were significantly older than the lay groups) perceived it often/regularly, and 36% perceived it sometimes (Table 1). The respective perception scores are significantly different (p < 0.0001). The perception that God is distant was experienced often or regularly in only a few cases and was experienced significantly less often in OLP. Further, the perception of being abandoned by God was experienced only in rare cases and less often in CLP (Table 1). Nevertheless, perceiving "spiritual emptiness" or inability to "give any more" was experienced significantly more often in RBS than in both lay person groups (Table 1).

Within the three samples, age was significantly and weakly associated with unanswered prayers in OLP only (Table 2). Within this group of OLP, the younger adults (<40 years of age) had the lowest

scores for unanswered prayers, while older persons (>60 years of age) had the highest (F = 3.2; *p* = 0.008). There were no gender-related differences (data not shown).

**Table 1.** Characterization of study persons. RBS: religious brothers and sisters; CLP: Catholic lay people; OLP: other lay people.

|  | **RBS** | **CLP** | **OLP** | *p*-**Value** |
|---|---|---|---|---|
| N | 273 | 716 | 351 | |
| Women/men (%) | 48.9/51.1 | 46.9/53.1 | 58.9/41.1 | 0.001 |
| Mean age (years) | 60.89 ± 13.54 | 54.84 ± 11.31 | 42.00 ± 16.54 | <0.0001 |
| **Prayers go unanswered (%)** | | | | |
|     Not at all | 20.4 | 35.4 | 35.1 | |
|     Seldom | 35.3 | 39 | 32.1 | |
|     Sometimes | 36.1 | 23 | 24.1 | <0.0001 |
|     Often | 5.9 | 3.2 | 6 | |
|     Regularly | 2.2 | 0.4 | 2.7 | |
| **God is distant (%)** | | | | |
|     Not at all | 39.4 | 37.9 | 53 | |
|     Seldom | 29.4 | 34.4 | 25.7 | |
|     Sometimes | 26 | 25.6 | 13.8 | <0.0001 |
|     Often | 3.3 | 2.1 | 6 | |
|     Regularly | 1.9 | 0 | 1.5 | |
| **Abandoned by God (%)** | | | | |
|     Not at all | 58.9 | 73.6 | 55.7 | |
|     Seldom | 25.2 | 19.7 | 30.7 | |
|     Sometimes | 13.7 | 5.9 | 11 | <0.0001 |
|     Often | 1.5 | 0.7 | 1.7 | |
|     Regularly | 0.7 | 0 | 0.9 | |
| **Spiritual Dryness (SDS)** | 2.28 ± 0.67 | 1.93 ± 0.63 | 1.95 ± 0.71 | <0.0001 |
|     Experience spiritual dryness | 2.74 ± 0.80 | 2.26 ± 0.86 | 2.26 ± 1.00 | <0.0001 |
|     Own prayers go unanswered | 2.34 ± 0.94 | 1.95 ± 0.87 | 2.09 ± 1.03 | <0.0001 |
|     God is distant | 1.99 ± 0.98 | 1.92 ± 0.85 | 1.77 ± 1.00 | 0.01 |
|     Abandoned by God | 1.60 ± 0.83 | 1.34 ± 0.62 | 1.61 ± 0.82 | <0.0001 |
|     Spiritually empty | 2.39 ± 0.94 | 1.90 ± 0.86 | 1.79 ± 0.90 | <0.0001 |
|     Not able to give any more | 2.59 ± 0.89 | 2.16 ± 0.95 | 2.17 ± 0.95 | <0.0001 |
| **Gratitude/Awe (GrAw-7)** | 72.17 ± 14.33 | 71.64 ± 17.67 | 68.58 ± 18.27 | n.s. |
| **Perception of the Sacred (DSES-6)** | 25.38 ± 5.55 * | - | 20.58 ± 7.49 | <0.0001 |
| **Well-being (WHO5)** | 15.42 ± 4.27 * | - | 15.38 ± 4.27 | n.s. |

\* *n* = 271.

**Table 2.** Correlation analyses with the three study samples.

|  | **Correlation Between "Unanswered Prayers" and Other Measures** | | |
|---|---|---|---|
|  | **RBS** | **CLP** | **OLP** |
| Spiritual Dryness (SDS) | **0.714 \*\*** | **0.762 \*\*** | **0.801 \*\*** |
|     Experience spiritual dryness | 0.426 ** | 0.435 ** | **0.547 \*\*** |
|     God is distant | 0.393 ** | **0.549 \*\*** | **0.636 \*\*** |
|     Abandoned by God | 0.386 ** | 0.430 ** | **0.547 \*\*** |
|     Spiritually empty | **0.524 \*\*** | **0.512 \*\*** | **0.543 \*\*** |
|     Not able to give any more | 0.352 ** | 0.442 ** | 0.313 ** |
| Well-being (WHO5) | −0.179 ** | - | −0.097 |
| Perception of the Sacred (DSES-6) | −0.381 ** | - | −0.086 |
| Gratitude/Awe (GrAw-7) | −0.180 [1] | −0.191 ** | −0.088 [2] |
| Frequency of private prayer | - | −0.162 ** | - |
| Age (years) | 0.013 | −0.052 | 0.204 * |

[1] *n* = 187; [2] *n* = 89; strong correlations are highlighted (bold). \*\* *p* < 0.01; \* *p* < 0.05 (Spearman rho).

### 3.3. Correlation Analyses

Within the three study samples, the correlation pattern between the perception that prayers go unanswered and other indicators of spiritual dryness showed similarities but also differences (Table 2). This perception was strongly associated with the experience of being "spiritually empty" and moderately with the perception of not being able to "give any more" in all three samples. The perception that God is distant was strongly related to unanswered prayers in both lay person samples, while it was only moderately related in RBS. In OLP, the perception that prayers go unanswered was strongly related to the experience of spiritual dryness and of being abandoned by God, while it was only moderately related in RBS and CLP.

Whether the perception of the sacred is related or not with the perception that prayers go unanswered was addressed in RBS and OLP. Interestingly, a moderate association was found in RBS only, while it was not significantly related in OLP (Table 2). Thus, although the perception of the sacred and awe/gratitude was strongly interrelated in RBS ($r = 0.501$) and moderately in OLP ($r = 0.418$), the association with the perception that prayers go unanswered was different.

Because we found that "unanswered prayers" were related to indicators of spiritual/emotional exhaustion and emptiness, we further analyzed their association with well-being. In OLP, the perception of not being able to give any more was moderately and inversely related ($r = -0.340$), but only marginally, in RBS ($r = -0.199$). In contrast, the perception of being "spiritually empty" was weakly negatively related to unanswered prayers in RBS ($r = -0.269$) and in OLP ($r = -0.237$). However, well-being was marginally negatively related to unanswered prayers in RBS, but not significantly related in OLP (Table 2).

Next, we analyzed the influence of gratitude/awe and frequency of private prayer on the perception of unanswered prayers and found that, in all cases, the associations were inversely and only marginally related (Table 2). Frequency of private prayer was marginally negatively related to the perception that prayers go unanswered in CLP (Table 2).

### 3.4. Predictors of Unanswered Prayers within the Three Samples

Next, we performed independent stepwise regression analyses within the three samples to reveal the best predictors of the perception that prayers go unanswered. In RBS, the perception of being "spiritually empty" was the best predictor, with further influence of feeling abandoned by God. This model explains 38% of variance (Table 3). Adding perception of the sacred (DSES-6) to the model increases the predictive power only weakly (adding 2% of explained variance).

**Table 3.** Regression analyses among RBS.

| Model 1: $R^2 = 0.382$ | | Beta | T | *p* |
|---|---|---|---|---|
| | (constant) | | 4.107 | <0.0001 |
| | God is distant | 0.075 | 1.187 | 0.236 |
| 1 | Abandoned by God | 0.183 | 2.942 | 0.004 |
| | Spiritually empty | 0.416 | 6.809 | <0.0001 |
| | Not able to give any more | 0.083 | 1.481 | 0.140 |
| **Model 2: $R^2 = 0.399$** | | | | |
| | (constant) | | 4.320 | <0.0001 |
| | God is distant | 0.054 | 0.850 | 0.396 |
| | Abandoned by God | 0.163 | 2.628 | 0.009 |
| 2 | Spiritually empty | 0.386 | 6.272 | <0.0001 |
| | Not able to give any more | 0.076 | 1.377 | 0.170 |
| | Perception of the Sacred | −0.145 | −2.682 | 0.008 |

In CLP, the perception that God is distant was the best predictor, with further feelings of being abandoned by God, being spiritually empty, and not being able to give any more (Table 4). These four variables explain 43% of variance.

**Table 4.** Regression analyses among CLP.

| Model 1: $R^2 = 0.428$ | | Beta | T | *p* |
|---|---|---|---|---|
| | (constant) | | 4.228 | <0.0001 |
| | God is distant | 0.298 | 8.110 | <0.0001 |
| 1 | Abandoned by God | 0.200 | 5.816 | <0.0001 |
| | Spiritually empty | 0.222 | 6.189 | <0.0001 |
| | Not able to give any more | 0.125 | 3.614 | <0.0001 |

In OLP, the perception that God is distant was the best predictor, with further feelings of being abandoned by God and being spiritually empty, while the perception of not being able to give any more was of relevance in trend only (Table 5). These variables explain 47% of variance. Adding perception of the sacred (DSES-6) to the model had no influence on the predictors.

**Table 5.** Regression analyses among OLP.

| Model 1: $R^2 = 0.474$ | | Beta | T | *p* |
|---|---|---|---|---|
| | (constant) | | 3.510 | 0.001 |
| | God is distant | 0.425 | 7.749 | <0.0001 |
| 1 | Abandoned by God | 0.193 | 3.850 | <0.0001 |
| | Spiritually empty | 0.148 | 2.793 | 0.006 |
| | Not able to give any more | 0.083 | 1.812 | 0.071 |
| **Model 2: $R^2 = 0.475$** | | | | |
| | (constant) | | 1.727 | 0.085 |
| | God is distant | 0.456 | 7.952 | <0.0001 |
| | Abandoned by God | 0.191 | 3.712 | <0.0001 |
| 2 | Spiritually empty | 0.145 | 2.696 | 0.007 |
| | Not able to give any more | 0.055 | 1.179 | 0.239 |
| | Perception of the sacred | 0.054 | 1.245 | 0.214 |

## 4. Discussion

We have chosen three different samples of persons to examine (1) the prevalence of perceiving that one's own prayers go unanswered and (2) which indicators of spiritual dryness predict this perception. The group of RBS is supposed to pray several times per day, while the group of CLP, who is responsible for presiding over the "Liturgy of the Word" on Sundays (or during the week) is supposed to pray frequently and regularly, but without an explicit demand. The important role of lay persons for the Sunday liturgy was discussed by Winter (2005). The group of OLP has no specific responsibilities in a religious community and served as a "control" group. However, the OLP was the youngest sample and scored the lowest on gratitude/awe (as a secular form of spirituality) and the lowest on perception of the sacred in daily life. They were assumed to be the "less religious" group. Interestingly, spiritual dryness was the highest in RBS as compared to both lay person groups.

The predictors of the perception of unanswered prayers were different in these three groups: For RBS, the best predictor was the perception of being "spiritually empty", indicating some kind of emotional/spiritual exhaustion. For CLP, the feeling that God is distant, feeling abandoned by God, and feeling "spiritually empty" were similarly relevant predictors. Here, it is a mix of perceived loss and distance and resulting emotional/spiritual exhaustion. In OLP, the perception that God is distant was the best predictor. This would indicate that, particularly in this (younger) group, spiritual doubt about whether God is responding at all is of particular relevance. In this group of OLP, the general

perception of the sacred in their life had no association with perceiving that prayers go unanswered, while perception of the sacred was higher in RBS and, for them, moderately negatively associated with prayers going unanswered. We wonder if the ability to perceive the sacred in life is higher in RBS because of their more focused lifestyle. This would also imply they more often have feelings of awe and gratitude than lay persons, and indeed, their gratitude/awe score was moderately to strongly correlated with perceiving the sacred in life. However, gratitude/awe was marginally or not at all related to perceptions of unanswered prayers. In conclusion, it was not an inability to be aware of sacred moments in life that was associated with the perception that prayers go unanswered but, rather, other reasons.

The predictors of perceived unanswered prayers varied most between RBS ("spiritual emptiness") and OLP (God is "distant"), while for the CLP, who have a duty in church, both predictors were relevant. How can this be explained? Well-being was similar in RBS and OLP, and age was of weak relevance only in OLP. In this group, age was of relevance particularly in younger adults, who may have other experiences in life as compared to older persons. Further, in particular, persons < 30 years (students) had the lowest perception of the sacred (mean 2.26 ± 1.34) as compared to older persons and the whole sample (mean 3.09 ± 1.50; F = 12.6; p < 0.0001), while gratitude/awe showed no age-related differences in OLP (F = 1.1; n.s.). Even when excluding OLP < 40 years from the sample, the best predictors of unanswered prayers in this group were the perception that God is distant (Beta = 0.476; T = 5.88; p < 0.0001) and being abandoned by God (Beta = 0.238; T = 3.11; *p* = 0.002). As a result, these observations cannot be easily explained by age and different life experiences alone. Further, while RBS and CLP were all Catholics, within the sample of OLP, 52% were Catholics, and 34% were Protestants, while 5% were of other religious denominations, and 8% had no specific religious denomination. However, perception of the sacred was similar in Catholics (20.94 ± 7.04) and Protestants (22.14 ± 7.09) and highest in the small group of persons of other religious denomination (25.54 ± 8.84) and very low in those without a specific religious denomination (14.34 ± 7.21); these differences were significant (F = 8.9; p < 0.0001). Further, the perception that prayers go unanswered was similar in Catholics (2.19 ± 1.03) and Protestants (2.09 ± 0.94) and lowest in those of other denominations (1.54 ± 0.66) and without (1.55 ± 1.06); these differences were significant (F = 4.1; *p* = 0.007). This may raise further questions about the perceived impact of prayer or its psychological and spiritual components on the perceptions and feelings of those who pray (Jors et al. 2015; Baumann 2016; Peng-Keller 2017).

Taking the experiential dimension of prayer seriously, prayers are not necessarily answered perceptively by God; such a presumption is not even the case among those who pray. The silence of God towards human prayer is a continuous topic in religious traditions, including the prayer experiences as expressed in the biblical psalms. Nevertheless, it is part of the experience of prayer that affects the praying person and generates feelings and insights that are perceived or rather interpreted as a result of a divine or transcendent responses to their prayers (Peng-Keller 2017). The interpretation that prayers go unanswered in contrast to other moments of life and prayer does affect the praying persons and groups in significantly different ways, as highlighted in the results presented above. There are both group factors and individual differences that determine how people deal with such experiences of spiritual dryness and frustration. The mere knowledge in religious traditions that such critical moments in prayer life are a call to faithfulness to one's spiritual commitment (Guardini 1998) can be of some help in working through these phases and gaining a more mature identity (Marcia 1993), after having explored more intensely what it actually means to pray.

## 5. Conclusions

Indicators of spiritual dryness, and particularly the experience that prayers go unanswered and that God is not responding, were significantly different in the three contrasting groups of religious brothers and sisters and lay persons. Emotional/spiritual exhaustion, on the one hand, and the perception that God is distant, on the other, were contrasting predictors of relevance. While the former was of relevance for all groups (and the best predictor in RBS), the latter was of particular relevance in

lay persons. This knowledge may help psychologists/psychotherapists, pastoral workers, and spiritual advisors to differentiate the underlying causes of spiritual dryness (in terms of "discernment") and thus support persons struggling with God, their faith, and life overall.

**Author Contributions:** Conceptualization and data analysis: A.B.; Writing-Original Draft Preparation, A.B. and K.B.; Funding Acquisition for CLP study: S.W. All authors have read and agreed to the published version of the manuscript.

**Funding:** Cross-sectional surveys among RBS and OLP received no external funding. The study among CLP was funded in part by German Conference of Bishops (DBK) and Rennings-Wagner-Stiftung.

**Acknowledgments:** 

**Conflicts of Interest:** A.B. is a university employee; S.W. is an employee of the Catholic diocese Osnabrück; K.B. is a Catholic priest and university employee. The authors declare no conflict of interest. The funding sponsors of the CLP study had no influence on data evaluation or interpretation, writing of the manuscript, or in the decision to publish the results.

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
