# Peer review of "Perception of Religious Brothers and Sisters and Lay Persons That Prayers Go Unanswered Is a Matter of Perceived Distance from God"

_religions, doi:10.3390/rel11040178_

Round 1

Reviewer 1 Report

This is a relevant topic for the field. Such research warrants attention in the social science discipline. However, are you merely writing this article for the religious community? If so, be clear.

Thus, it would have been useful for the authors identify the target audience(s)who would benefit from this study's findings. Including implications in the article would help to enhance the quality of the paper as you speak to the target audience identified in the intro who would find these findings meaningful in practical contexts.

Author Response

Thanks for the advise. Of course, several disciplines might have interst and we therefore tried to avoid a specific audience. Nevertheless, we will add the following sentence at the end of the Research Questions:

"This knowledge might be of relevance not only theoretically (i.e., theology, psychology  and social science), but also practically for pastoral workers, spiritual advisors and psychologists /psychotherapists."

Reviewer 2 Report

Perception that own prayers go unanswered in religious brothers and sisters and lay persons is a matter of perceived distance from God. Paper analyzes connections between prayer life and ‘spiritual dryness’ among three groups (religious brothers and sisters; Catholic lay persons; other lay persons). In the paper, clear methodological path to the results is presented. Reader is provided with many interesting results, eg. “…spiritual dryness was highest in RBS as comparted to both lay person groups.“

While reading the paper, three aspects kept occupying my mind:

  1. How these results link to extrinsic/intrinsic religiosity?
  2. What really is an unanswered prayer? This is quite a philosophical question and I could read few lines more on the topic in introduction section. From theological perspective, it could even be argued that there is no such thing – unanswered prayers then become peoples’ unwillingness to accept the will of God/higher power/transcendent (just to show the point that it is not easy topic).
  3. How does the results resonate with mature faith? As a term “mature identity”, is mentioned in the end of the discussion; yet, these results are also linked to mature faith/religiosity.

As the paper is coherent as such these three points are given only to illustrate what went through my mind. Yet, there might be something there to scope in a bit deeper if the author(s) wishes to do so.

Author Response

Thanks for the good comments!

  1. How these results link to extrinsic/intrinsic religiosity?

>> Good Point. At least for the first 2 groups we can assume that intrinsic religioisity is in the forefront. We gave a small hin only in the fist part, but have not investigated this in Detail and thus cannot discuss this substantially.

  1. What really is an unanswered prayer? This is quite a philosophical question and I could read few lines more on the topic in introduction section. From theological perspective, it could even be argued that there is no such thing – unanswered prayers then become peoples’ unwillingness to accept the will of God/higher power/transcendent (just to show the point that it is not easy topic).

>>From a theological point you are of course right. However, from an experiential point the situation might be different. It is of course a matter of 'egoistic' expectations that the "private Jesus" is not at hand to fulfill all my wishes. A a-theistic interpretation would be that there is not God to respond to prayers and that is an unidirectional communication. - We have added the following in the Introduction: "Of course it is true that the question whether the prayers are directly ‘answered’ at all is first of all a matter of expectation. Praying means to be in contact with God, to communicate with Him. Whether and how the ‘responses’ are perceived, is a different question. Either one may not notice the ‘response’ (which is still given), or the expectations are very specific how God should respond, or one expects an immediate response which might not be given. It is also true that this specific expecting form of praying is an attempt to control God. A different form is meditative or reflecting prayer which means to be with God, without any specific expectations, and be in resonance whatever may appear as a response – even when it is quietness and silence which may nevertheless calm the soul (and might be an answer to)."      

  1. How does the results resonate with mature faith? As a term “mature identity”, is mentioned in the end of the discussion; yet, these results are also linked to mature faith/religiosity.

>> The acceptance that God is the 'utmost other' is part of it, also that the respnses might be different than expected. - Having faith despite of this, reduced to the 'naked' trust in Garden Gethsemane God. 

Thanks for these comments!